# ACTIVA: Amortized Causal Effect Estimation via Variational Autoencoders

## Abstract

Predicting the distribution of outcomes under hypothetical interventions is crucial in healthcare, economics, and policy-making. However, existing methods often require restrictive assumptions, and are typically limited by the lack of amortization across problem instances. We propose ACTIVA, a transformer-based conditional variational autoencoder (VAE) architecture for amortized causal inference, which estimates interventional distributions directly from observational data. ACTIVA learns a latent representation conditioned on observational inputs and intervention queries, enabling zero-shot inference by amortizing causal knowledge from diverse training scenarios. We provide theoretical insights showing that ACTIVA predicts interventional distributions as mixtures over observationally equivalent causal models. Empirical evaluations on synthetic and semi-synthetic datasets validate our insights and show the effectiveness of our amortized approach, highlighting promising directions for future real-world applications.

## 1 Introduction

Understanding the causal effects of an action by pure observation is central in many domains including healthcare (Shi & Norgeot, 2022), economics (Panizza & Presbitero, 2014), and finance (Kumar et al., 2023). Yet, estimating these effects is constrained by inherent limitations regarding the conclusions that can be drawn from observations only (Bareinboim et al., 2022). Specifically, looking only at observational data leads to ambiguities over which causal effect a certain action will have.

To address these ambiguities, current approaches to causal inference often rely on strong assumptions(see 7 for more detail) to derive a specific point estimate of the effect. An alternative approach to address these ambiguities is to make them explicit in the form of uncertainties. For example, providing an interval or distribution of plausible causal effects has been shown to be feasible and useful in practice (Maathuis et al., 2009; Balazadeh et al., 2025). Such a prediction allows for a clearer picture of what effects to expect and can lead to actionable inferences if e.g. the range of effects lies within an interval that leads to the same actions.

Although having a distribution over causal effects is a promising direction, it only represents uncertainty over this specific point-estimate. A natural extension of this is to look at the overall distribution shift of the causal factors after performing an action. Additionally to answering cause-effect questions, such a distribution can provide further task-relevant insights such as multimodality and hence prevent misleading average effect predictions.

Recent work further emphasizes the reuse of learned knowledge of diverse causal tasks across different queries to improve inference efficiency by learning one model for the estimation of different problem instances (Löwe et al., 2022; Lorch et al., 2022; Scetbon et al., 2024; Sauter et al., 2024b; Mahajan et al., 2024; Annadani et al., 2025; Robertson et al., 2025; Balazadeh et al., 2025; Ma et al., 2025). Results from these approaches suggest that we can effectively amortize over datasets coming from different causal models for various downstream tasks.

In this paper, we propose a causal encoder-decoder architecture called ACTIVA. During training, the encoder maps the joint interventional and observational data to a latent space conditioned on observational data and an interventional query, and the decoder transforms the encoded causal information into an interventional distribution. At inference time, we condition our prior on the observational data alone to obtain a distribution over plausible causal models, which the decoder transforms into

a mixture over these models. We provide a theoretical analysis on the specific nature of our latent and decoded space. In empirical evaluations, we validate an implementation of ACTIVA that can successfully recover causal information at inference time even on novel instances and when some theoretical assumptions are violated. This allows for the potential zero-shot transfer from simulated scenarios to the real-world ones without knowing the causal relations, avoiding common pitfalls when relying on such graphs (Poinsot et al., 2024). We summarize our main contributions as follows:

- We provide a conditional variational autoencoder model (ACTIVA) designed for amortized causal distribution shift estimation. Our model takes an observational dataset and a query intervention as input, predicts a latent representation of the dataset and outputs an estimate of the respective interventional distribution.

- We analyze the proposed model theoretically, leading to rare fundamental insights about amortized causal inference. The result emphasizes that amortized causal inference is inherently limited by identifiability from the data. Nevertheless it can recover shift distributions up to a mixture of shifts from observationally equivalent models.

- We implement an instantiation of ACTIVA and show empirically that it predicts interventional distributions given observational data and interventional queries both on synthetic and semi-synthetic data, thus validating our theoretical model.

Overall, our work highlights the significance of amortized causal inference as a tool to overcome traditional hurdles in distributional causal effect estimation. The code to our model is available at `https://anonymous.4open.science/r/Amortized_Interventional_Distribution_Estimation-C90B/README.md`

## 2 BACKGROUND AND NOTATION

**Conditional $\beta$-VAEs** Variational Autoencoders (VAEs) are generative models that approximate the joint distribution $p_\theta(\boldsymbol{x}, \boldsymbol{z}) = p_\gamma(\boldsymbol{x}|\boldsymbol{z})p_\eta(\boldsymbol{z})$, where $\boldsymbol{z}$ are latent variables governing the data $\boldsymbol{x}$ and $\theta = \{\gamma, \eta\}$ are the parameters determining the data generation. The marginal data likelihood $p_\theta(\boldsymbol{x})$ is optimized using the evidence lower bound (ELBO) (Kingma & Welling, 2013):

$$\log p_\theta(\boldsymbol{x}) \geq \mathbb{E}_{q_\phi(\boldsymbol{z}|\boldsymbol{x})} \left[\log p_\gamma(\boldsymbol{x}|\boldsymbol{z})\right] - \mathrm{KL}(q_\phi(\boldsymbol{z}|\boldsymbol{x})\|p_\eta(\boldsymbol{z})). \tag{1}$$

Where $\phi$ parametrizes the encoding distribution $q$. The ELBO has a reconstruction term, which ensures the model accurately reconstructs the data from the latent representation, and a Kullback-Leibler divergence (KL) term, which regularizes the encoding distribution $q_\phi(\boldsymbol{z} \mid \boldsymbol{x})$ to approximate some prior $p_\eta(\boldsymbol{z})$.

Conditional VAEs (CVAEs) (Sohn et al., 2015) extend this framework to allow for conditional generation with some auxiliary information $\boldsymbol{c}$. The generative process of CVAEs follows a conditional prior such that

$$p_\theta(\boldsymbol{x}, \boldsymbol{z}|\boldsymbol{c}) = p_\gamma(\boldsymbol{x}|\boldsymbol{z}, \boldsymbol{c})p_\eta(\boldsymbol{z}|\boldsymbol{c}). \tag{2}$$

In $\beta$-VAEs (Higgins et al., 2017), the KL term is scaled by a hyperparameter $\beta$, leading to a modified ELBO as follows:

$$\mathbb{E}_{q_\phi(\boldsymbol{z}|\boldsymbol{x},\boldsymbol{c})} \left[\log p_\gamma(\boldsymbol{x}|\boldsymbol{z}, \boldsymbol{c})\right] - \beta \, \mathrm{KL}(q_\phi(\boldsymbol{z}|\boldsymbol{x}, \boldsymbol{c})\|p_\eta(\boldsymbol{z}|\boldsymbol{c})). \tag{3}$$

The choice of $\beta$ governs the trade-off between accurately reconstructing the data and disentangling the latent representations. While $\beta > 1$ emphasizes disentanglement, in our setting, we use $\beta < 1$ to prioritize accurate reconstruction.

**Causal Models** In this paper, we employ the notation of structural causal models (SCM) for describing causal data generating processes. For a detailed definition, we refer the reader to (Bareinboim et al., 2022).

We treat a causal model $\mathcal{M}$ as a generative process over $d$ variables $\boldsymbol{V} = \{V_1, ..., V_d\}$, denoting an assignment of these variables as $\boldsymbol{v} = \{v_1, ..., v_d\}$. The model is causal in the sense that each variable is the direct cause of a subset of $\boldsymbol{V}$. Specifically, any variable $V_j \in \boldsymbol{V}$ is determined by its

direct causes with $V_j \leftarrow f_j(Pa_{V_j}, U_j)$, where $f_j$ is an arbitrary function of the direct causes $Pa_{V_j}$ (representing parent variables of $V_j$) and a noise term $U_j$. We call two models equivalent, writing $\mathcal{M} = \mathcal{M}'$, if all their parameters are equal. All causal relations combined form the causal graph $G^{\mathcal{M}}$. Each model $\mathcal{M}$ induces a joint distribution $p_{\mathcal{M}}(\boldsymbol{V})$, called the observational distribution. All models with the same observational distribution form a so-called Markov equivalence class (MEC).

In a causal model, performing a so-called intervention $do(V = v)$ manipulates $\mathcal{M}$ such that the target variable $V$ is forced to take on the value $v$, regardless of $V$'s causes. Such an intervention results in an intervened model that we denote $\mathcal{M}_{do(V=v)}$ or $\mathcal{M}_{do(V)}$ when $v$ is clear from the context. We denote the variables of $\mathcal{M}_{do(V=v)}$ as $\boldsymbol{V} \mid do(V = v)$. In this work, $v$ is one possible intervention value for which the training data contains samples. Similarly to its observational counterpart, $\mathcal{M}_{do(V)}$ induces a joint distribution $p_{\mathcal{M}}(\boldsymbol{V} \mid do(V))$ that we call the interventional distribution.

The aim of this work is to estimate the causally shifted distribution $p_{\mathcal{M}}(\boldsymbol{V} \mid do(V))$ from an observational dataset $\boldsymbol{D}^{\mathcal{M}} = \boldsymbol{v}_{1:N}^{\mathcal{M}}$, where $N$ is the number of i.i.d. samples and $\boldsymbol{v}_n^{\mathcal{M}} \sim p_{\mathcal{M}}(\boldsymbol{V})$. In general, identifying this distribution is not possible without additional assumptions or interventional data because of the ambiguities introduced by MECs (Bareinboim et al., 2022). This fundamental property of causality naturally extends to our work, which we address in the theoretical portion of this work on data with identifiable effects.

## 3 ACTIVA

In this section, we outline a $\beta$-CVAE architecture for amortized causal distribution shift estimation called ACTIVA. In our setup, a dataset of observational samples and a query intervention serve as a conditional for the following generative process:

$$p_\theta(\boldsymbol{V}, \boldsymbol{z} \mid \boldsymbol{D}^{\mathcal{M}}, do(V)) = p_\gamma(\boldsymbol{V} \mid \boldsymbol{z}) p_\eta(\boldsymbol{z} \mid \boldsymbol{D}^{\mathcal{M}}, do(V)) \tag{4}$$

Intuitively, the data is modeled by a conditional prior that maps the observational data $\boldsymbol{D}^{\mathcal{M}}$ and query intervention $do(V)$ to a latent code $\boldsymbol{z}$. The latents then contain all the information the model needs to reconstruct the data; in our case, the samples from a post-interventional distribution $\boldsymbol{V}$.

To find parameters $\theta$, we use the conditional ELBO formulation for likelihood maximization as in equation 3. Translated to our model, our overall learning objective becomes:

$$\mathcal{L}(\theta, \phi) = \underset{q_\phi(\boldsymbol{z} \mid \boldsymbol{V}, \boldsymbol{D}^{\mathcal{M}}, do(V))}{\mathbb{E}} [\log p_\gamma(\boldsymbol{V} \mid \boldsymbol{z})] - \mathrm{KL}(q_\phi(\boldsymbol{z} \mid \boldsymbol{V}, \boldsymbol{D}^{\mathcal{M}}, do(V)) \parallel p_\eta(\boldsymbol{z} \mid \boldsymbol{D}^{\mathcal{M}}, do(V))),$$
$$\tag{5}$$

where $q_\phi(\boldsymbol{z} \mid \boldsymbol{V}, \boldsymbol{D}^{\mathcal{M}}, do(V))$ is the encoding distribution, $p_\gamma(\boldsymbol{V} \mid \boldsymbol{z})$ the decoder distribution, and $p_\eta(\boldsymbol{z} \mid \boldsymbol{D}^{\mathcal{M}}, do(V))$ the conditional prior.

To estimate the objective from data, we assume to have training data $\mathbf{D}^{tr} = [\boldsymbol{v}_n^{\mathcal{M}_{do(V)}}, \boldsymbol{D}^{\mathcal{M}}, do(V)]_{j=0}^{J}$ with $J$ training examples pairing interventional samples with observational ones and the corresponding intervention. Where $\mathbf{D}_j^{tr} \sim p_{tr}(\mathbf{D}^{tr}) = \int p_{tr}(\mathbf{D}^{tr} \mid \mathcal{M}) p_{tr}(\mathcal{M}) d\mathcal{M}$ is an element from the training data distribution according to a pre-defined distribution of causal models $p_{tr}(\mathcal{M})$. This pre-defined distribution can be seen as the simulator that models causal models that we expect at inference time.

To enable gradient-based optimization, we use the reparameterization trick (Kingma & Welling, 2013), allowing backpropagation through the stochastic sampling process of $q_\phi$. For our final objective, we aim at amortizing prediction over the training distribution of the causal model, to successfully apply our model to data sets that were not in the training data. Optimizing our amortized objective then amounts to minimizing the expectation of our loss regarding the distribution of the training datasets.

$$\min_{\theta, \phi} \underset{\mathbf{D}_j^{tr} \sim p_{tr}(\mathbf{D}^{tr})}{\mathbb{E}} \mathcal{L}(\theta, \phi) \tag{6}$$

This objective ensures that our model can make inferences on datasets coming from the class of pre-defined causal models, even if the specific dataset was not in the training set. It furthermore implies that the more general the training distribution of data sets is, the more the model will generalize to new data sets during inference, as has also been argued in (Montagna et al., 2024).

## 4 THEORETICAL ANALYSIS

In this section, we show that inference with the model specified above gives us a mixture of all causal shift distributions that are consistent with the observational data. We start with some assumptions:

**Assumption 1:** In the infinite sample limit, each causal model $\mathcal{M}_j$ is identifiable by the data it generated $\mathbf{D}_j^{tr} = [\boldsymbol{v}_n^{\mathcal{M}_{do(V)}}, \boldsymbol{D}^{\mathcal{M}}, do(V)]_j$ in the training data.

This means that we rely on the identifiability of the model from the joint interventional and observational data samples we use for training. In practice, we rely on amortization on (semi-) synthetic data for training for which this assumption is easily satisfied by performing more interventions or considering causal models that are identifiable even from purely observational data.

**Assumption 2:** Under Assumption 1, $q_\phi(\boldsymbol{z}|\boldsymbol{V}, \boldsymbol{D}^{\mathcal{M}}, do(V))$ identifies the causal model, i.e., $q_\phi(\boldsymbol{z}|\mathbf{D}_j^{tr}) = q_\phi(\boldsymbol{z}|\mathbf{D}_i^{tr}) \Rightarrow \mathcal{M}_j = \mathcal{M}_i$.

In other words, we assume that our parametrized variational posterior $q_\phi$ maps each data point to a distinct latent distribution.

**Assumption 3:** The decoder is injective, i.e., for two inputs $\boldsymbol{z} \neq \boldsymbol{z}' \Rightarrow dec_\gamma(\boldsymbol{z}) \neq dec_\gamma(\boldsymbol{z}') \Rightarrow p_\gamma(\boldsymbol{V}|\boldsymbol{z}) \neq p_\gamma(\boldsymbol{V}|\boldsymbol{z}')$.

While this assumption can be satisfied with various architectures (Rezende & Mohamed, 2015; Kingma et al., 2016), we show in our experiments that even when the decoder is not invertible, our approach remains effective.

**Lemma 1:** Under Assumptions 1-3, $q_\phi(\boldsymbol{z}|\boldsymbol{V}, \boldsymbol{D}^{\mathcal{M}}, do(V))$ is a Dirac distribution.

*Proof:* Take any two equivalent models $\mathcal{M} = \mathcal{M}'$ and any two samples $z \sim q_\phi(\boldsymbol{z}|\boldsymbol{V}, \boldsymbol{D}^{\mathcal{M}}, do(V))$, $z' \sim q_\phi(\boldsymbol{z}|\boldsymbol{V}, \boldsymbol{D}^{\mathcal{M}'}, do(V))$. Suppose that the two samples differ, i.e. $z \neq z'$, then we know from Assumption 3 that $p_\gamma(\boldsymbol{V}|\boldsymbol{z}) \neq p_\gamma(\boldsymbol{V}|\boldsymbol{z}')$. But since we know that they represent the same model (Assumption 2), the two resulting distributions over $\boldsymbol{V}$ must be the same, which leads to a contradiction. Therefore, $z$ and $z'$ must be the same. Accordingly, $q_\phi(\boldsymbol{z}|\boldsymbol{V}, \boldsymbol{D}^{\mathcal{M}}, do(V))$ must always yield the same sample, and the proof is complete. $\square$

Together with Assumption 2, this allows us to interpret $z_k$ as the unique latent representation of the causal model $\mathcal{M}^k$ and intervention query. Note that directly using Dirac distributions is impractical due to undefined gradients. In our implementation we, therefore, approximate Dirac distributions with multivariate normal (MVN) distributions, using a fixed, small variance and zero covariance.

Next, we investigate what the learned prior will look like under the above assumptions. Our analysis relies on re-arranging the training data in the loss computation and the ELBO following Hoffman & Johnson (2016). More specifically, we define $\mathbf{D}_{[\mathcal{M}^o]}^{tr}$ as the set of training examples within the Markov equivalence class with representative $\mathcal{M}^o$. In other words, $\mathbf{D}_{[\mathcal{M}^o]}^{tr}$ contains all data-points where the probability of generating such observational data-points by $\mathcal{M}$ is the same as the one by $\mathcal{M}^o$. Furthermore, we define $\bar{q}_\phi(\boldsymbol{z}|\mathbf{D}_{[\mathcal{M}^o]}^{tr}, do(V)) := \frac{1}{|[\mathcal{M}^o]|} \sum_{\mathbf{D}_j^{tr} \in \mathbf{D}_{[\mathcal{M}^o]}^{tr}} q_\phi(\boldsymbol{z}|\mathbf{D}_j^{tr})$ which is the distribution that averages over all encoder distributions that identify models that are observationally equivalent to $\boldsymbol{D}^{\mathcal{M}^o}$. Finally, this allows us to derive from equation 5, the only term affecting our prior is

$$\mathrm{KL}(\bar{q}_\phi(\boldsymbol{z}|\mathbf{D}_{[\mathcal{M}^o]}^{tr}, do(V))||p_\eta(\boldsymbol{z}|[\boldsymbol{D}^{\mathcal{M}}, do(V)]_o)). \tag{7}$$

The detailed derivation can be found in Appendix A.1. To ensure the KL is well-defined, we additionally make the following assumption.

**Assumption 4:** The prior is a finite mixture of Dirac distributions $p_\eta(\boldsymbol{z}|\boldsymbol{D}^{\mathcal{M}}, do(V)) = \sum_C \rho_c \delta_{\boldsymbol{z}_c}(z)$ and is absolutely continuous with respect to $\bar{q}_\phi$.

Intuitively, absolute continuity here implies that we have prior knowledge on which latents do not fall into the observational equivalence class of the input dataset. While this seems like a strict restriction, we point out that in practice we implement the prior as a Gaussian mixture model (GMM), hence it satisfies this assumption naturally and without prior knowledge on the equivalence classes.

**Proposition 1:** Under Assumption 1-4, the prior $p_\eta(\boldsymbol{z}|[\boldsymbol{D}^{\mathcal{M}}, do(V)]_o)$ equals $\bar{q}_\phi(\boldsymbol{z}|\mathbf{D}_{[\mathcal{M}^o]}^{tr}, do(V)) \forall o \in O$ when the ELBO is minimized.

*Proof:* Firstly, note that the KL term in equation 7 needs to become zero to be minimized for each $o$. As per Assumption 4 and the definition of $\bar{q}_\phi$, this term becomes $\sum_C \pi_c \log \frac{\pi_c}{\rho_c}$ which is 0 exactly when $\forall c : \pi_c = \rho_c = \frac{1}{|[\mathcal{M}^o]|}$. Hence, $p_\eta(\boldsymbol{z}|[\boldsymbol{D}^{\mathcal{M}}, do(V)]_o) = \bar{q}_\phi(\boldsymbol{z}|\mathbf{D}^{tr}_{[\mathcal{M}^o]}, do(V))$ when equation 6 is minimized. $\square$

Proposition 1 shows that the prior that we learn through optimization of equation 6 corresponds exactly to a uniform distribution over the latent points defining the corresponding observational equivalence class, providing an explicit notion of uncertainty over the true causal model. This uncertainty propagates to the estimated post-interventional distribution as we show in the following.

**Proposition 2:** The conditional generative distribution $p_\theta(\boldsymbol{V}|[\boldsymbol{D}^{\mathcal{M}}, do(V)]_j)$ is a mixture of distributions $\sum_{\mathcal{M}^k \in [\mathcal{M}^j]} \rho_k p(\boldsymbol{V}|\boldsymbol{D}^{\mathcal{M}^k}, [do(V)]_j)$.

*Proof:* To show this, we analyze the marginal $p_\theta(\boldsymbol{V}|[\boldsymbol{D}^{\mathcal{M}}, do(V)]_j)$ of equation 4. Because of Lemma 1 and Proposition 1 we can write the marginal as a sum over the latents $\boldsymbol{z}_k \in [\boldsymbol{z}_j]$ that represent models $\mathcal{M}^k \in [\mathcal{M}^j]$ yielding $\sum_{[\boldsymbol{z}_j]} p_\gamma(\boldsymbol{V}|\boldsymbol{z}_k)p_\eta(\boldsymbol{z}_k|[\boldsymbol{D}^{\mathcal{M}}, do(V)]_j)$. Since $p_\eta$ is a uniform mixture and $\boldsymbol{z}_k$ encodes the model-query pair, this is equivalent to $\sum_{[\boldsymbol{z}_j]} \rho_k p_\gamma(\boldsymbol{V}|\boldsymbol{z}_k) = \sum_{\mathcal{M}^k \in [\mathcal{M}^j]} \rho_k p_\theta(\boldsymbol{V}|\boldsymbol{D}^{\mathcal{M}_k}, [do(V)]_j)$. $\square$

This shows that ACTIVA estimates the causal distribution shift up to a mixture of observationally equivalent models. Our predicted distributions explicitly encode uncertainty about the data-generating causal model. Furthermore, these results are, to the best of our knowledge, alongside results from Balazadeh et al. (2025) , among the first steps to build a theory around amortized causal inference. Figure 1 shows the intuition behind our result.

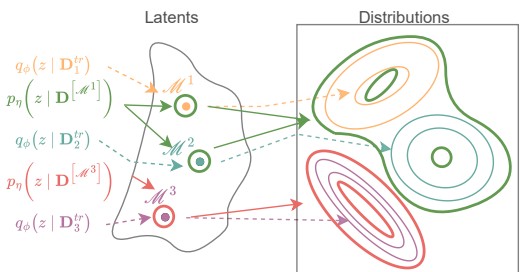

Figure 1: Given observational data $\boldsymbol{D}^{\mathcal{M}}$ and a query intervention $do(X = x)$, the encoder maps to a latent point and the prior to a mixture of latent points. Samples $\boldsymbol{z}$ from this latent space are passed to the decoder, which estimates the interventional distribution $p_{\mathcal{M}}(\boldsymbol{V} \mid do(X = x))$ as a Gaussian mixture. Dashed lines indicate training-time information flow, solid lines represent inference-time.

## 5 MODEL ARCHITECTURE

**Representing Interventions** To represent interventions, we create a matrix representation of the interventions. We consider $i \in \{1, \ldots, |I|\}$ the index of a possible intervention value $v_i$, where $|I|$ is the number of possible values, and a selector $\boldsymbol{t} \in \{0, 1\}^d$ indicating the target(s) of the intervention. By not encoding the intervention value directly, but rather the index of the intervention, we remove the need to have a full specification of the intervention. This allows us to model interventional distributions based on placeholder IDs of the interventions, as long as they can be attributed to interventional training data and have specified targets.

We construct the intervention representation $\boldsymbol{I}_{i,t}$ representing $do(\boldsymbol{V_t} = v_i)$ as follows. We perform a one-hot encoding of $i$ creating a vector $\boldsymbol{i_{oh}} \in \mathbb{R}^{|I|}$ and repeat it $d$ times, resulting in a matrix $\boldsymbol{i_{rep}} \in \mathbb{R}^{d \times |I|}$. We then apply $\boldsymbol{t}$ as a mask to this matrix, effectively zeroing out the rows that correspond to non-intervened variables. Finally, we repeat the intervention representation $N$ times to match the number of input samples and obtain $\boldsymbol{I}_{i,t} \in \mathbb{R}^{N \times d \times |I|}$.

This construction ensures that the intervention-relevant information for each variable is provided as local information alongside the variable itself and maintains permutation equivariance.

**Embedding Network** We encode the input data and intervention according to an embedding network $h_\alpha(\boldsymbol{D}^{\mathcal{M}}, \boldsymbol{I}_{i,t})$ based on the extension of non-parametric encoders (Kossen et al., 2021; Lorch et al., 2022). For various causal tasks, this architecture can successfully encode a dataset into a vector containing causally relevant information, such as structure (Lorch et al., 2022), topological ordering (Scetbon et al., 2024), informative interventions (Annadani et al., 2025), and effects (Balazadeh et al., 2025; Robertson et al., 2025; Ma et al., 2025).

Given a dataset $\boldsymbol{D}^{\mathcal{M}} \in \mathbb{R}^{N \times d}$ of $N$ observational samples from a model $\mathcal{M}$, an intervention index $i$ and a binary target vector $\boldsymbol{t}$, we append $\boldsymbol{I}_{i,t}$ to the dataset resulting in an augmented dataset $\boldsymbol{D}_{it}^{\mathcal{M}} \in \mathbb{R}^{N \times d \times |I|+1}$. Then we apply $L$ blocks of multi-head self-attention (MHSA) that alternate in attending over $d$ features and $N$ samples. Furthermore, we apply layer normalization, residual connections, and dropout following the transformer setup (Vaswani et al., 2017). After the transformer blocks, we average over the sample axis to obtain an embedding $\boldsymbol{h} \in \mathbb{R}^{d \times e}$, where $e$ indicates the embedding dimension, thus ensuring that the embedding is permutation invariant regarding the sample dimension and equivariant regarding the feature dimension.

**Prior and Encoder** We choose the conditional prior to be an n-dimensional GMM with constant variance and no covariance whose means, and weights are determined by a learnable prior function $[\boldsymbol{\mu}_{0:d}, \rho_{0:C}] = pri_\eta(h_{\alpha'}(\boldsymbol{D}^{\mathcal{M}}, \boldsymbol{I}_{i,t}))$. Specifically, $pri_\eta$ are two two-layer multi-layer perceptron (MLP) that we apply to the embedding $\boldsymbol{h}'$ to each feature independently to compute the means and mixture weights. The outputs parametrize our prior distribution $p_\eta(\boldsymbol{z}|\boldsymbol{D}^{\mathcal{M}}, do(V)) = \sum_{j=0}^{|C|} \rho_j \mathcal{MVN}(\boldsymbol{z}; \boldsymbol{\mu_j}, 0.1^C)$, with and $\boldsymbol{\mu_j}$ the mean vector of each component, respectively. We enforce $\sum_{j=0}^{C} \rho_j = 1$ via a softmax layer.

Furthermore, we choose the encoder distribution to be an n-dimensional MVN with a constant, diagonal covariance matrix whose means are determined by a learnable encoder function $\boldsymbol{\mu} = enc_\phi(h_{\alpha''}(\boldsymbol{V}, \boldsymbol{I}_{i,t}), h_{\alpha'}(\boldsymbol{D}^{\mathcal{M}}, \boldsymbol{I}_{i,t}))$, where $h_{\alpha'}$ is the observational data network used in the prior, and $h_{\alpha''}$ is the network encoding interventional data with different parameters. We concatenate the resulting embeddings $\boldsymbol{h}'$ and $\boldsymbol{h}''$ along the embedding dimensions and again apply a two-layer MLP to each feature independently to compute the mean. This fully parametrizes our encoding distribution $q_\phi(\boldsymbol{z}|\boldsymbol{V}, \boldsymbol{D}^{\mathcal{M}}, do(V)) = \mathcal{MVN}(\boldsymbol{z}; \boldsymbol{\mu}, 0.1)$.

**Decoder** We model the decoder $dec_\gamma(\boldsymbol{z})$ as a transformer that outputs the parameters of a Gaussian mixture, enabling a closed-form expression for the estimated interventional distributions.

For the decoder we process $\boldsymbol{z}$ via $K$ standard transformer blocks (Vaswani et al., 2017) resulting in the embedding $\boldsymbol{h}_{dec}$. We feed $\boldsymbol{h}_{dec}$ through a row-wise linear layer to predict the means $\boldsymbol{\mu}_\gamma \in \mathbb{R}^{B \times d}$, where $B$ is the number of decoder mixture components, maintaining permutation equivariance regarding the variable ordering. To also predict the covariances $\boldsymbol{\Sigma}_\gamma \in \mathbb{R}^{B \times d \times d}$ in a permutation equivariant manner, we first apply a row-wise linear layer to compute $\boldsymbol{u} \in \mathbb{R}^{B \times d \times e}$ as an intermediate step and then compute the covariances via $\boldsymbol{u} \cdot \boldsymbol{u}$ similarly to (Lorch et al., 2022). Lastly, we predict the component weights $\boldsymbol{b}_{0:B}$ of the mixture by first summing $\boldsymbol{h}_{dec}$ along the $d$ feature dimensions and then passing the pooled representation through a linear layer and a softmax layer.

We note that we maintain permutation equivariance of the means and covariances of the resulting mixture w.r.t. the ordering of the variables, a property that has been shown to be important in causal inference for scaling (Lorch et al., 2022) and accuracy (Li et al., 2020), among others.

# 6 EXPERIMENTS

We evaluate the performance of the proposed method across three different types of datasets: two purely synthetic and one semi-synthetic dataset. Below we provide an overview of each dataset category. Detailed information on dataset generation can be found in Appendix A.2.

## 6.1 EXPERIMENTAL SETUP

**Synthetic Data (Gaussian and Beta Noise).** We generate data from linear additive causal models of the form $V_j = \sum_{i \in \mathrm{Pa}(j)} \beta_{ij} V_i + \varepsilon_j$, where $\varepsilon_j$ is drawn either from a Gaussian distribution $\mathcal{N}(0, \sigma^2)$ or a Beta distribution $\mathrm{Beta}(\alpha, \beta)$. We generate data for single-variable interventions $do(V_i = 5)$ on each variable. In general, linear Gaussian-noise models are not identifiable without additional constraints or interventional data, whereas beta-noise models are identifiable from merely observational data (Peters et al., 2017), which makes these two classes interesting for comparison.

**Semi-Synthetic (SERGIO).** We generate biologically inspired data using the SERGIO simulator (Dibaeinia & Sinha, 2020) for gene expression. SERGIO models single-cell gene expressions with the resulting data aligning closely with real gene expression patterns. Notably, we use the imple-

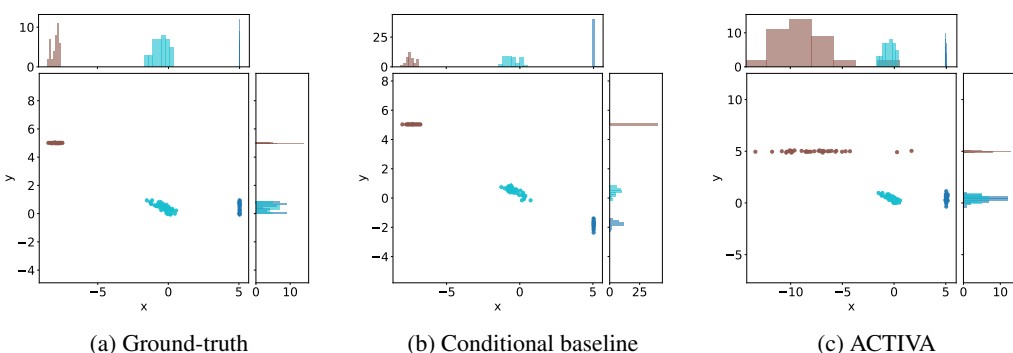

(a) Ground-truth          (b) Conditional baseline          (c) ACTIVA

Figure 2: Samples from the observational and interventional distributions of the ground-truth model (a), the conditional baseline (b) and our model (c) for the *Beta* data. The turquoise represent samples from the observational distribution, blue and brown depict samples from the interventional distributions by setting $X$ and $Y$ to 5, respectively. Note that for visualization purposes, we provide the observational samples of the original data for the ACTIVA plot.

mentation provided by (Lorch et al., 2022) that allows interventions in the simulator. We generate data for single-variable interventions $do(V_i = 0)$ on each variable to simulate gene knockout.

**Metrics.** To assess the fidelity of the learned distributions, we rely on the Maximum Mean Discrepancy (MMD) (Gretton et al., 2012), the Wasserstein Distance (WSD) (Villani, 2009) and the Energy Distance (ERG) (Székely & Rizzo, 2013) to get a broad picture of the distributional accuracy of our estimations. For all three metrics, lower values are desired. A detailed introduction to these metrics can be found in Appendix A.3.

**Conditional Baseline.** We implement a baseline that approximates the interventional distribution $p(V \mid do(V_t = v_i))$ using conditional distributions. It fits a multivariate normal (MVN) to the dataset to approximate the observational distribution $p(V)$. From this MVN, it derives the conditional distribution $p(V \setminus V_t \mid V_t = v_i)$. Then, it draws $N$ samples from this conditional distribution and assigns the intervention value $v_i$ to each variable in $V_t$. This approach enables us to compare the degree to which our method captures causal information, rather than relying solely on conditioning.

**Inference.** To infer the causal distribution shift from our trained model, we follow the following procedure. First, the observational data and interventional query are provided to the prior as input, resulting in the prior distribution over latents. Since the prior is a GMM, obtaining closed-form solutions for interventional distributions at inference is intractable. We instead approximate the distribution by sampling 50 latent variables. We then decode each of these latent samples and form a uniform mixture of the resulting decoded distributions. Since our metrics are all sample-based, we then sample from this distribution.

## 6.2 DISTRIBUTION SHIFT ESTIMATION

In this experiment, we examine whether our model's inference procedure indeed captures causal distribution shifts. We train our model on the *Beta* and *Gaussian* datasets described above. We first consider the bivariate case where each dataset consists of pairs of variables $(X, Y)$ under distinct causal mechanisms. Next, we extend this evaluation by training ACTIVA on *Gaussian* and *Beta* datasets with 8 variables to show its ability to scale to more complex scenarios. Details about hyperparameter settings are presented in Appendix A.4.

**Quantitative Analysis.** We assess our method's performance by comparing it against the conditional baseline on a test set of novel causal models. The models are novel in the sense that while they come from the same distribution of models $p_{tr}(\mathcal{M})$, the models that were sampled in the test set are not in the training set. Table 1 reports the average scores on the two test sets in the bivariate and 8 variable case, performing inference with one sample from the prior.

Table 1: Average inference performance of our trained model and the baseline on held-out test data with 2 and 8 variables respectively. The *Data* column indicates which dataset has been used with how many variables.

| Data | ACTIVA | | | Baseline | | |
| --- | --- | --- | --- | --- | --- | --- |
| | MMD | WSD | ERG | MMD | WSD | ERG |
| Gauss 2 | 0.29 | 1.9 | 2.0 | 0.91 | 8.0 | 7.9 |
| Beta 2 | 0.21 | 1.4 | 1.4 | 0.90 | 5.8 | 11.0 |
| Gauss 8 | 0.65 | 11.4 | 12.9 | 0.88 | 19.9 | 16.4 |
| Beta 8 | 0.77 | 9.3 | 13.7 | 0.94 | 11.2 | 19.0 |
| SERGIO | 0.34 | 4.3 | 1.7 | 0.50 | 3.9 | 2.8 |
| OOD | 0.37 | 7.6 | 3.7 | 0.61 | 8.0 | 8.0 |

We observe that ACTIVA outperforms the baseline across all metrics and for both model sizes. This shows that our implementation of ACTIVA can recover causal distribution shifts beyond conditioning. Furthermore, we observe in the 2 variable case the increased performance on the Beta data compared to the Gaussian data. We interpret this as reflecting the identifiability of the Beta data compared to the general non-identifiability of the Gaussian data. This result validates our main proposition as the observational equivalence classes for the Beta data must be smaller than for the Gaussian case and hence the resulting mixture prediction more precise. Lastly, we note that our approach yields strong performance even for the Gaussian case, which is known to be non-identifiable in general.

**Qualitative Analysis.** Figure 2 provides a qualitative view of the learned models on the held-out test *Beta* dataset. In this example, there is a negative causal effect from $Y$ to $X$. We observe that both the baseline and ACTIVA recover the mean in the causal direction (brown). As discussed before, the higher uncertainty of ACTIVA in itself can provide a useful signal for practical application, as this uncertainty, potentially over multiple causal models, is explicit and can be exploited. In the anticausal direction (blue), the baseline predicts a distribution shift while ACTIVA correctly predicts *no* causal effect.

These results indicate that our method successfully leverages causal information during inference. By performing a single forward pass, the model estimates accurate interventional distributions that respect the true causal mechanisms governing the data-generating process and makes uncertainties explicit.

### 6.3 PERFORMANCE ON SEMI-SYNTHETIC DATA

To highlight the potential application of ACTIVA to real scenarios, we employ the *SERGIO* (Dibaeinia & Sinha, 2020) simulator for generating gene expression data. Following the setup in (Lorch et al., 2022), we create training and test sets with 8 variables, as well as an out-of-distribution (OOD) evaluation dataset with 11 variables. Appendix A.2.2 provides a detailed list of the used parameters.

Table 1 compares the baseline to ACTIVA trained on the SERGIO data on the test set and the OOD evaluation set, reporting the average performance metrics. The results largely align with the ones found in Section 6.2, with the exception of almost equal performance on the WSD.

We also observe that performance degrades on the OOD data for both ACTIVA and the baseline. However, in addition to reflecting more challenging conditions than the train data, this decrease is partly confounded by the fact that the metrics are sensitive to data scale as detailed in A.3. Hence, changes in scale between the test (mean value 2.16) and OOD dataset (mean value 3.71) can amplify discrepancies in the reported metrics, even without a fundamental reduction in model performance.

We conclude that ACTIVA exhibits robust performance on unseen semi-synthetic data, confirming its applicability to more realistic domains. Furthermore, ACTIVA remains effective even under distribution shifts, highlighting its potential for handling OOD scenarios. Its advantage over the conditional baseline on semi-synthetic OOD data further highlights the benefits of its causal archi-

tecture for estimating interventional distributions and its potential transfer from simulated training data to real-world test data.

## 7 RELATED WORK

Deep learning approaches to causal inference have advanced modeling of observational, interventional, and counterfactual distributions, yet most demand known or learnable causal graphs and retraining for each new dataset. This holds for variational-autoencoder–based models (Yang et al., 2021; Qi & Yu, 2023), diffusion generative models (Sanchez et al., 2022; Chao et al., 2023), graph-neural and flow-based approaches (Zečevi et al., 2021; Sánchez-Martin et al., 2022; Poinsot et al., 2024), adversarial network approaches (Rahman & Kocaoglu, 2024).

Recent amortized estimators leverage prior-fitted networks or supervised learning to predict causal effect distributions without additional training or known causal graphs (Balazadeh et al., 2025; Robertson et al., 2025; Ma et al., 2025). However, they still rely on explicit covariate specification and typically target predefined estimands. Other meta-learning frameworks are restricted to supervised effect estimation (Bynum et al., 2025), enabling zero-shot estimation for unseen interventions. (Nilforoshan et al., 2023), model the noise as latent (Pawlowski et al., 2020; De et al., 2023) within the same causal model context, require upfront model assumptions (Löwe et al., 2022) and graphs (Mahajan et al., 2024).

Joint inference methods that jointly learn graphs and mechanisms likewise require dataset-specific optimisation (Lorch et al., 2021; Nishikawa-Toomey et al., 2022; Deleu et al., 2023). Other approaches focus on estimating treatment-outcome relationships without modeling the whole distribution (Louizos et al., 2017; Vowels et al., 2021; Wu & Fukumizu, 2023), require dataset-specific retraining (Khemakhem et al., 2021; Melnychuk et al., 2023) or are limited to specific classes of causal models (Cundy et al., 2021). Some approaches focus on special cases, such as limited overlap or marginal interventional data, but cannot handle joint distributions across all variables (Vanderschueren et al., 2023; Garrido et al., 2024).

In contrast, our method amortizes across both causal structures and mechanisms, requiring only observational data and a specified intervention to return the full joint interventional distribution in a single forward pass. It therefore removes the need for retraining, covariate decomposition, or graphical assumptions.

## 8 CONCLUSION

We introduce ACTIVA, a VAE architecture for amortized causal distribution estimation. During training, the encoder maps interventional data — conditioned on observational inputs and a query intervention — to a latent representation, and the decoder reconstructs the corresponding interventional distribution. At inference time, conditioning solely on observational data yields a prior over plausible causal models, which the decoder transforms into a mixture of model-specific interventional distributions. We provide a rigorous theoretical analysis of these latent and output spaces and empirically validate our implementation on synthetic and semi-synthetic datasets. ACTIVA's primary strengths include its capability for zero-shot inference using observational data alone. This work is among the first works that offer a theoretical foundation for amortized causal inference.

Future work will focus on bridging the gap between theoretical assumptions and implementation, thoroughly evaluating ACTIVA's performance on real-world datasets to better understand the simulation-to-reality transfer, and exploring alternative deep learning architectures aligned with the developing theory of amortized causal inference. We hope this work inspires further developments in rethinking causal inference in an amortized learning context with both theoretical and practical advances.

## REPRODUCIBILITY STATEMENT

We took several steps to support reproducibility. An anonymous repository with training and evaluation code, and instructions is provided in the supplementary materials (see the code link in the main

text). Our model details and architectural choices are specified in Section 5, with the theoretical assumptions and proofs discussed in detail in Section 4. Complete dataset generation procedures (synthetic and SERGIO), splits (including OOD), and intervention settings are documented in Appendix A.2; evaluation metrics and their implementations are described in Appendix A.3; and all optimization settings, hyperparameters, seeds, and hardware are summarized in Appendix A.4. We also describe baselines and inference procedures within Section 6 to facilitate like-for-like comparisons. Lastly, we provide the datasets needed to reproduce some of our experiments.

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

# A APPENDIX

## A.1 DERIVATION OF MARGINAL KL

In this Section we further detail how we derive the KL divergence term presented in Section 4. We start by rearranging the amortized loss computation in equation 6 as follows by sorting the training samples accordingly: $\frac{1}{J} \sum_{\mathbf{D}^{tr}} \mathcal{L} = \frac{1}{J} \sum_I \sum_O \sum_{\mathbf{D}^{tr}_{[\mathcal{M}^o]}} \mathcal{L}$, where $J$ is the number of data points, $I$ is the set of possible interventions, $O$ is a partition over the datasets according to their Markov equivalence class and $\mathbf{D}^{tr}_{[\mathcal{M}^o]}$ is the set of training examples within the equivalence class with representative $\mathcal{M}^o$, s.t. $\mathbf{D}^{tr}_{[\mathcal{M}^o]} = \{[\boldsymbol{v}_n^{\mathcal{M}_{do(V)}}, \boldsymbol{D}^{\mathcal{M}}]_j \in \mathbf{D}^{tr} : p_{\mathcal{M}}(\boldsymbol{D}^{\mathcal{M}}) = p_{\mathcal{M}^o}(\boldsymbol{D}^{\mathcal{M}^o})\}$. In other words, $\mathbf{D}^{tr}_{[\mathcal{M}^o]}$ contains all data points where the probability of generating such observational data points by $\mathcal{M}$ is the same as the one by $\mathcal{M}^o$. Such a rearrangement is possible since it constitutes a simple reordering of the input samples and each sample can only be in one observational equivalence class at once.

Rearranging the training samples in this manner is crucial for alignment with the formulation of the alternative ELBO by Hoffman & Johnson (2016) as they marginalize over the posterior input while keeping the prior input the same. We define in the following $\bar{q}_\phi(\boldsymbol{z}|\mathbf{D}^{tr}_{[\mathcal{M}^o]}, do(V)) := \frac{1}{|[\mathcal{M}^o]|} \sum_{\mathbf{D}^{tr}_j \in \mathbf{D}^{tr}_{[\mathcal{M}^o]}} q_\phi(\boldsymbol{z}|\mathbf{D}^{tr}_j)$ which is a uniform mixture of Dirac distributions as per Lemma 1 with $\pi_c = \frac{1}{|[\mathcal{M}^o]|}$ for all component weights $\pi_c$ by construction. In other words, $\bar{q}_\phi$ is the distribution that averages over all encoder distributions that identify models that are observationally equivalent to $\boldsymbol{D}^{\mathcal{M}^o}$. Treating the sample index $j$ as random variable, we can further define $q(j, \boldsymbol{z}) := \frac{1}{|[\mathcal{M}^o]|} q(\boldsymbol{z}|\mathbf{D}^{tr}_j)$. Finally, this allows us to decompose the KL term in our loss into

$$\frac{1}{|[\mathcal{M}^o]|} \sum_{\mathbf{D}^{tr}_{[\mathcal{M}^o]}} \mathrm{KL}(q_\phi(\boldsymbol{z}|\mathbf{D}^{tr}_j) \| p_\eta(\boldsymbol{z}|[\boldsymbol{D}^{\mathcal{M}}, do(V)]_j))$$

$$= \mathrm{KL}(\bar{q}_\phi(\boldsymbol{z}|\mathbf{D}^{tr}_{[\mathcal{M}^o]}, do(V))\|p_\eta(\boldsymbol{z}|[\boldsymbol{D}^{\mathcal{M}}, do(V)]_o)) + \mathbb{I}_{q(j,\boldsymbol{z})}[j, \boldsymbol{z}], \quad (8)$$

where $\mathbb{I}_{q(j,\boldsymbol{z})}[j, \boldsymbol{z}]$ is the mutual information. In the main text, we then focus on only the KL term since it is the only one affecting the prior parameters.

## A.2 DATASETS

We evaluate the performance of the proposed method across three different types of datasets: two purely synthetic (*Gaussian* and *Beta*) and one semi-synthetic (*SERGIO*). Below, we provide an overview of each dataset category along with the relevant parameters. For every causal model in these categories, we draw $N$ samples from both the observational and the interventional distributions to create paired datasets. Details about the exact number of models, how we split into training/test sets, and further parameter choices are given in the subsections below.

### A.2.1 SYNTHETIC DATA (GAUSSIAN AND BETA NOISE)

We generate data from linear additive causal models of the form:

$$V_j = \sum_{i \in \mathrm{Pa}(j)} \beta_{ij} V_i + \varepsilon_j,$$

where $\varepsilon_j$ is drawn from either: A Gaussian distribution, $\mathcal{N}(0, \sigma^2)$, or A Beta distribution, $\mathrm{Beta}(\alpha, \beta)$. For each variable $V_i$, we generate interventional data by applying a single-variable intervention $do(V_i = 5)$ plus some noise $\mathcal{N}(0, 0.1)$ to increase numerical stability. In total, this yields one observational dataset and $d$ interventional datasets for each causal model. All synthetic data is generated with the *Causal Playground* library (Sauter et al., 2024a).

The causal graphs are generated according to the ER procedure (Erdós & Rényi, 1959) where first we generate an ER graph with edge-probability 0.3 and then randomly remove edges until the graph becomes acyclic. When the noise terms follow a Gaussian distribution, we sample $\beta_{ij}$ uniformly from $[-2, -0.5] \cup [0.5, 2]$ and let $\varepsilon_j \sim \mathcal{N}(0, 0.5^2)$. For the Beta case, we again draw $\beta_{ij}$ in $[-2, -0.5] \cup [0.5, 2]$, but the noise terms $\varepsilon_j$ follow $\mathrm{Beta}(\alpha, \beta)$ with $\alpha, \beta \sim \mathrm{Uniform}(0.5, 2)$.

Table 2: SERGIO data generation parameters.

| parameter | in-distribution | out-of-distribution |
|---|---|---|
| genes | 8 | 11 |
| b | Uniform(0, 1) | Uniform(0.5, 2.0) |
| k_param | Uniform(1, 5) | Uniform(3, 7) |
| k_sign | Beta(1, 1) | Beta(0.5, 0.5) |
| hill | $\in\{$ 1.9, 2.0, 2.1 $\}$ | $\in\{$ 1.5, 2.5 $\}$ |
| decays | $\in\{$ 0.7, 0.8, 0.9 $\}$ | $\in\{$ 0.5, 1.5 $\}$ |
| noise_params | $\in\{$0.9, 1.0, 1.1$\}$ | $\in\{$0.5, 1.5$\}$ |

We examine both a 2-variable and an 8-variable scenario:

- **Two Variables.** We randomly generate 10000 linear models and sample 40 points each for the observational and interventional distributions. We split these 10000 models into 9000 for training and 1000 for testing.

- **Eight Variables.** We generate 30000 linear models. For each model, we sample 30 data points for the observational and for each interventional distribution. Of these 30000 models, 27000 are used for training and 3000 are held out for testing.

A.2.2  SEMI-SYNTHETIC DATA (SERGIO)

To evaluate on data with biological realism, we employ the SERGIO simulator (Dibaeinia & Sinha, 2020), which generates single-cell gene-expression data. Notably, we use the version of SERGIO provided in (Lorch et al., 2022), which includes functionality for performing interventions. We treat single-gene knockouts as interventions, thus applying $\mathrm{do}(V_i = 0)$ to each gene $V_i$. This yields one observational dataset and 8 single-gene interventional datasets per simulated gene-regulatory network.

**Simulator Settings and Network Structures.** Following the procedure in (Marbach et al., 2009; Lorch et al., 2021), we randomly sample subgraphs of known gene-regulatory networks from *E. coli* or *S. cerevisiae*, ensuring that each subgraph has 8 genes. For each subgraph, we draw model parameters (e.g., activation constants, decay rates, and noise magnitudes) from predefined ranges (Table 2. We then generate observational data as well as data from each gene-knockout intervention.

**Training, Testing, and Out-of-Distribution (OOD) Splits.** We run 15,000 SERGIO simulations for our in-distribution dataset and sample 30 cells (data points) for each observational and interventional condition. We then split these 15,000 simulations into training (90%) and test (10%). We also construct an OOD set of 1800 simulations, where some SERGIO parameters (e.g., noise or decay rates) are sampled from partially disjoint ranges, creating a controlled distribution shift. For evaluation, we again sample 30 cells per observational/interventional condition in these OOD simulations.

In summary, the synthetic datasets allow us to systematically analyze the ability of our amortized approach to capture linear causal effects under distinct noise assumptions, while the semi-synthetic SERGIO datasets bring our method closer to real-world conditions by simulating biologically realistic gene-expression data under interventions.

A.3  METRICS

To quantitatively assess how closely our estimated interventional distributions align with the ground-truth or observed distributions, we employ three sample-based distance measures and a permutation-based statistical test. Specifically, we use:

- **Maximum Mean Discrepancy (MMD)**

- **Wasserstein Distance (WSD)**

- **Energy Distance (ERG)**

Each of these distances captures a distinct notion of distributional similarity. The MMD is a kernel-based embedding measure that detects differences in distribution means in a reproducing kernel Hilbert space; the Wasserstein distance considers a geometric "transport cost" viewpoint; and the energy distance focuses on pairwise distance comparisons; Taken together, these three metrics give a robust picture of how well our learned distributions match the ground truth in terms of shape, location, and higher-order moments.

**Maximum Mean Discrepancy (MMD).** The MMD (Gretton et al., 2012) compares the mean embeddings of two distributions, $P$ and $Q$, in a reproducing kernel Hilbert space (RKHS). Concretely, for samples $\{\mathbf{x}_i\}_{i=1}^m \sim P$ and $\{\mathbf{y}_j\}_{j=1}^n \sim Q$, MMD is computed as:

$$\mathrm{MMD}^2(P, Q) \;=\; \mathbb{E}_{\mathbf{x}, \mathbf{x}' \sim P}[k(\mathbf{x}, \mathbf{x}')] \;+\; \mathbb{E}_{\mathbf{y}, \mathbf{y}' \sim Q}[k(\mathbf{y}, \mathbf{y}')] \;-\; 2\,\mathbb{E}_{\mathbf{x} \sim P, \mathbf{y} \sim Q}[k(\mathbf{x}, \mathbf{y})],$$

where $k(\cdot, \cdot)$ is a positive-definite kernel. Lower MMD values indicate that the two sets of samples are more similar with respect to that kernel-induced feature map. In our experiments, we use a Gaussian (RBF) kernel

$$k_\sigma(\mathbf{x}, \mathbf{y}) = \exp\left(-\frac{\|\mathbf{x} - \mathbf{y}\|^2}{2\sigma^2}\right),$$

whose bandwidth parameter $\sigma$ is chosen via the median heuristic (i.e., by setting the kernel scale to the median distance among all pairs of training samples). This approach is a common, adaptive way to select an appropriate kernel width without extensive hyperparameter tuning (Garreau et al., 2017).

**Wasserstein Distance (WSD).** Also referred to as the Earth Mover's Distance, the Wasserstein distance (Villani, 2009) evaluates how much "effort" is needed to transform one distribution into another. Formally, for $p \geq 1$:

$$\mathcal{W}_p(P, Q) \;=\; \left(\inf_{\gamma \in \Gamma(P,Q)} \int_{\mathcal{X} \times \mathcal{X}} d(\mathbf{x}, \mathbf{y})^p \, d\gamma(\mathbf{x}, \mathbf{y})\right)^{1/p},$$

where $\Gamma(P, Q)$ is the set of all couplings of $P$ and $Q$, and $d(\mathbf{x}, \mathbf{y})$ is a distance metric (usually Euclidean). A lower Wasserstein distance indicates that the two distributions are closer in a geometric sense, reflecting both differences in location and in spread.

**Energy Distance (ERG).** The energy distance (Székely & Rizzo, 2013) offers yet another perspective on distributional similarity by comparing pairwise distances:

$$\mathrm{ED}(P, Q) \;=\; \mathbb{E}\big[\|\mathbf{X} - \mathbf{Y}\|\big] \;-\; \tfrac{1}{2}\,\mathbb{E}\big[\|\mathbf{X} - \mathbf{X}'\|\big] \;-\; \tfrac{1}{2}\,\mathbb{E}\big[\|\mathbf{Y} - \mathbf{Y}'\|\big],$$

where $\mathbf{X}, \mathbf{X}' \sim P$ and $\mathbf{Y}, \mathbf{Y}' \sim Q$. Lower energy distance values indicate higher overall similarity between $P$ and $Q$ in terms of both mean locations and dispersion patterns.

**Energy-Based Permutation Test.** Finally, we apply a permutation test based on the energy distance to determine whether two sets of samples are statistically indistinguishable from one another. Specifically, we first compute the observed energy distance $D_{\mathrm{obs}}$ between the real (or ground-truth) samples and the model-generated samples. Then we pool the two sets of samples and repeatedly sample 100 random permutations to split them into two groups of the original sizes. Finally we compute the energy distance $D_{\mathrm{perm}}$ for each permutation, thereby approximating a null distribution in which $P$ and $Q$ are "mixed."

If $D_{\mathrm{obs}}$ is not significantly larger than typical $D_{\mathrm{perm}}$ values (at a chosen significance level), we fail to reject the null hypothesis that both sets of samples come from the same distribution. Hence, a high $p$-value indicates that our learned samples are statistically indistinguishable from the ground truth.

## A.4 Hyperparameters

Table 3 provides a summary of the principal hyperparameters used in our experiments, along with their chosen values for each model variant.

**Models and Experiments.**

- *Gauss 2* and *Beta 2* refer to the models used in the bivariate experiments described in 6.2.

- *Gauss 8* and *Beta 8* analogously represent versions for the eight-variable models.
- *SERGIO* is the version of our model employed for the semi-synthetic gene-expression data of the experiment in 6.3).

**Explanation of Table Columns.**

- **Batch size** indicates how many datasets are processed in each training step.
- **Heads** is the number of heads used in the multi-head self-attention layers.
- **K** (in our table notation) specifies the number of Transformer blocks used in the decoder.
- **L** is the number of Transformer blocks in the encoder, alternating attention over sample and feature axes. This always has to be an even number.
- **Dropout** is the probability of randomly dropping units in the attention and feed-forward sublayers.
- **e** indicates the embedding dimension for each feature before attention is applied.
- **h** hidden dimension of the neural networks throughout the model.
- **d** is the dimensionality of input features.
- **c** denotes the number of Gaussian components used in the mixture for the decoder's output distribution.
- **k** denotes the number of Gaussian components used in the mixture for the prior distribution.
- **epochs** is the total number of passes through the training dataset.
- **lr** is the learning rate used for the Adam-based optimizer (Dozat, 2016).
- **seed** ensures reproducibility of parameter initialization and dataset splits.
- $\beta$ is the coefficient that balances the KL term in our $\beta$-VAE training objective (see Section 4 of the main text). A value less than 1 places more emphasis on accurate reconstruction over disentanglement.
- **GPU** on which the model was trained.

Table 3: Hyperparameter configurations for the main experiments. Each row represents one of our trained models, indicating how it was set up in terms of architecture depth, dimensionality, and optimization parameters.

|  | batch size | heads | K | dropout | e | h | d | c | k | L | epochs | lr | seed | $\beta$ | GPU |
|---|---|---|---|---|---|---|---|---|---|---|---|---|---|---|---|
| **Gauss 2** | 256 | 2 | 4 | 0.1 | 64 | 128 | 2 | 1 | 2 | 2 | 5000 | 0.001 | 42 | 0.1 | A6000 |
| **Beta 2** | 256 | 2 | 4 | 0.1 | 64 | 128 | 2 | 1 | 2 | 2 | 5000 | 0.001 | 42 | 0.1 | A6000 |
| **Gauss 8** | 256 | 4 | 4 | 0.1 | 64 | 512 | 2 | 1 | 2 | 4 | 3600 | 0.0005 | 42 | 0.1 | A6000 |
| **Beta 8** | 256 | 4 | 4 | 0.1 | 64 | 512 | 2 | 1 | 2 | 4 | 3600 | 0.0005 | 42 | 0.1 | A6000 |
| **SERGIO** | 256 | 4 | 4 | 0.1 | 64 | 256 | 2 | 1 | 2 | 4 | 4100 | 0.0005 | 42 | 0.1 | A100 |

Training times vary depending on model size ($d$, $e$, $c$) and dataset size (batch size, epochs). Larger mixture components ($c$) and higher embedding dimensions ($e$) generally increase both training time and representational capacity. Potential NaN's in the optimization are zeroed. All modules and training scripts were implemented with Jax (Bradbury et al., 2018) and Flax (Heek et al., 2024). Experiment were run on the DAS6 computing cluster (Bal et al., 2016).

In summary, the hyperparameters in Table 3 reflect our balancing of model complexity, regularization, and computational resource constraints. Values are chosen empirically to ensure stable training and satisfactory performance across different experimental settings.

## A.5 LLM USAGE

Throughout this work, we made use of various large language models (LLMs) as general-purpose assistive tools across different stages of the research process. Specifically, LLMs were employed to support literature exploration, clarify technical formulations, check the consistency of proofs, assist

with coding and debugging, suggest concise rephrasings for improved readability, and aid with formatting tasks (e.g., LaTeX adjustments). All scientific contributions, conceptual developments, and final claims remain the responsibility of the authors.

