# OpenReview forum: "ACTIVA: Amortized Causal Effect Estimation via Variational Autoencoders"
_ICLR.cc/2026/Conference — Submitted to ICLR 2026_

### Official Review · Reviewer_4cjR · 2025-10-26

**Soundness:** 2
**Presentation:** 2
**Contribution:** 2
**Rating:** 2
**Confidence:** 4

**Summary:**

This paper proposes ACTIVA (Amortized Causal Effect Estimation via Variational Autoencoders), a Transformer-based conditional β-VAE framework designed to perform amortized causal inference. The goal is to learn a generalizable model that can estimate interventional distributions p(V∣do(X)) directly from observational data, enabling zero-shot causal reasoning on unseen causal models. Theoretical results claim that ACTIVA approximates a mixture over all Markov-equivalent causal models, and empirical results on synthetic and semi-synthetic datasets show improved metrics compared to conditional baselines.

**Strengths:**

1. The work tackles a timely and relevant problem: enabling amortized or foundation-style causal inference that generalizes across tasks.
2. Empirical results suggest that the proposed method improves over simple conditional baselines in terms of distributional similarity metrics.

**Weaknesses:**

1. The proposed method essentially combines a conditional VAE with a Transformer backbone for dataset-to-distribution mapping. The main innovation claimed “amortized causal inference” has been explored in prior works such as Neural Causal Models, MetaCausal Inference, and Do-PFN. The paper lacks a clear articulation of what distinguishes ACTIVA from these approaches, beyond architectural variations.
2. The theoretical results (Propositions 1–2) mainly formalize that the learned distribution corresponds to a mixture over Markov-equivalent models. This is not new and does not directly establish identifiability or consistency guarantees. The connection between the ELBO optimization and causal identifiability remains mostly qualitative.
3. The experiments are restricted to low-dimensional synthetic datasets. There is no evaluation on real or large-scale causal datasets, nor comparison to recent amortized or foundation causal inference baselines. As a result, the empirical validation is insufficient to support the ambitious claims of generalizable causal inference.
4. The model outputs are distributions, but there is no qualitative or quantitative analysis of which causal mechanisms have been captured or whether the learned latent space aligns with meaningful causal factors. Without such interpretability checks, the claim that ACTIVA “amortizes causal reasoning” remains speculative.

**Questions:**

See Weaknesses Part.

---

> ### Author Response · Authors · 2025-11-21
>
> We appreciate your comments.
>
> 1. Neural Causal Models (assuming Xia et al. 2022) does as far as we understand not treat the amortized setting, but rather a per-instance setting. For MetaCausal inference we would like to ask the reveiwer  to provide us with the full reference so we can give an appropriate answer. Do-PFN (Robertson et al. 2025) does indeed solve a very similar task, but assumes known covariates at inference, as we point out in the related work section.
>
> 2. Indeed, our theoretical results characterize the kind of distributions ACTIVA learns and explicitly does not introduce new identifiability results. We do believe though, that the theoretical characterization we provide for the amortized conditional VAE setting in causal inference i) underlines the soundness and intuition behind our approaches and ii) fills a gap for such models and tasks in the field in general.
>
> 3. For the comparison with further baselines, please see the reply to reviewer U8Ab. Regarding the critique about evaluations, this is a common issue  in this field in general. So in our case, even after thoroughly searching for it , we could not find real-world datasets that fit the setup of our method. We welcome and would appreciate any pointers at this point, should you be aware of such data\simulator.
>
> 4. We provide a qualitative example in the paper that shows the capture of the correct causal and anticausal direction. Quantitatively, you are suggesting an interesting follow-up investigation. We agree that it would indeed be interesting what causal mechanisms are captured in the latent. Our  hypothesis here would be that even the causal graph g should be recoverable from a latent point z. We do argue though, that showing the encoding of such causal factors in our latent is not a necessary condition to make claims about amortized causal reasoning. We believe that showing distributional match already provides a strong evidence that causal mechanisms were captured. Indeed our comparison to the baseline aims at exactly this argument: we argue that the match would be significantly worse if the estimated mixture distribution would not follow causal rules.
>
> We hope that these explanations have resolved the questions and concerns in your review.

---

### Official Review · Reviewer_hs82 · 2025-10-30

**Soundness:** 4
**Presentation:** 3
**Contribution:** 4
**Rating:** 8
**Confidence:** 3

**Summary:**

The paper presents ACTIVA (Amortized Causal Inference via Variational Autoencoding), a framework for amortized estimation of interventional distributions. Given an observational dataset $D_M$ and a query intervention $do(V)$, it outputs an estimate of the post-intervention joint distribution $p(V\mid do(V))$ . Experiments on synthetic Gaussian/Beta causal systems and semi-synthetic SERGIO data show improved fidelity to ground-truth interventional distributions compared to a conditional VAE baseline.

**Strengths:**

- The paper is clearly written and well-structured.
- Assumptions, lemmas, and propositions are clearly stated.
- Consistent improvements on all metrics (MMD, Wasserstein, Energy) across multiple data regimes, with good OOD generalization.

**Weaknesses:**

- The amortization relies on paired interventional + observational data from a simulator / training distribution; real-world acquisition of such pairs may be costly, and generalization hinges on simulator-to-real match.
- The focus is entirely on distributional distances and the empirical experiments lack evaluation on downstream causal quantities (e.g., ATE / ITE).
- Limited statistical analysis as no confidence intervals / paired tests are provided.

**Questions:**

1. How sensitive is ACTIVA to misspecification of the training model family ($p_{tr}(M)$)?
2. How sensitive are results to the number of GMM components and mixture size in the decoder? Would increasing mixture components improve uncertainty calibration?
3. Please include CIs / paired tests for Table 1.

---

> ### Author Response · Authors · 2025-11-21
>
> We thank the reviewer for the useful comments and questions. We also appreciate the positive receptions.
>
> - We would like to emphasize that paired data of observations and interventions is only needed for the training of our model, which as you say, can be simulated. During inference, our model does not rely on interventional data for inference. This aims exactly at alleviating the difficulty of finding such data pairings in the real-world.
>
> - The question of downstream point estimates touches on a nuanced behaviour of our model. Specifically, we believe that given the prediction of a mixture of distributions as in our approach, point estimates like ATE might occlude the actual behaviour. To be more explicit, imagine an example where our model estimates the mixture over two Gaussian distributions (representing the causal queries on the observationally equivalent models), then we have an estimated distribution with two modes. Now we argue that taking the expected value of such distribution does not give us  useful  information about the two underlying true effects.
> We are happy to extend future versions of this paper with more thorough statistical analysis.
>
> - Q1: We expect some degree  of robustness to out-of-distributions just as our experiments with the SERGIO OOD data indicates. In particular, we think that much of the structural information from one specification can be generalized to another one. For the misspecification of model parameters and functional relations we expect bigger differences.
>
> - Q2: In preliminary experiments, we did not encounter  any significant difference  in performance with more decoder components. We attribute this observation to the fact that our datasets follow uni-modal distributions. We would expect  better uncertainty with more decoder components for multimodal and/or non-Gaussian data distributions.
>
> - Q3: Please see the previous answer.
>
> We hope these clarifications have sufficiently addressed the questions and concerns raised in your review.

---

### Official Review · Reviewer_RehM · 2025-11-04

**Soundness:** 1
**Presentation:** 2
**Contribution:** 1
**Rating:** 2
**Confidence:** 4

**Summary:**

This paper proposes a method for causal inference using variational autoencoders (VAEs), interpreting the prior and decoder distributions as a mixture over causal models. It claims to enable zero-shot transfer to unknown causal settings by training on a predefined class of identifiable models $\mathcal{M}$, allowing inference on novel datasets from $\mathcal{M}$ without prior knowledge of specific causal relations. The approach leverages strong identifiability assumptions (e.g., full model recovery and injectivity) and a mixture prior to amortize inference across instances. Theoretical justification is provided. Experiments are done on synthetic linear additive models and a semi-synthetic dataset

**Strengths:**

- The perspective of interpreting the prior and generative (decoder) distributions as a mixture over causal models is a conceptually appealing idea.
- The work pursues an ambitious research direction: enabling zero-shot transfer to unknown causal settings, which could advance causal discovery and inference in generative models.

**Weaknesses:**

**Overly Strong Assumptions That Trivialize the Problem**
The core assumptions render the problem overly simplistic.
- Assumption 1 identifies the entire causal model, which far exceeds the identification of treatment effects (TE) alone. This is unnecessarily restrictive.
- Assumption 2 effectively sidesteps the central problem by presupposing full identifiability from the posterior. Identifiability in VAEs remains an open research area (see [1, 2] and "Missing Related Work" below).

**Incorrect Theoretical Results and Derivations**

- Equation (4) posits that $V$ is independent of $D^M$ and $\mathrm{do}(V)$ given $z$. Independence from $D^M$ stems from the strong requirement that the latent $z$ "contains all the information needed to reconstruct the data," which should be explicitly stated as an assumption. However, independence from $\mathrm{do}(V)$ is ill-formulated: intervened variables are, by definition, dependent on (and determined by) the intervention, typically modeled via a Dirac delta. All subsequent derivations must account for this. Moreover, implied by this independence, $z$ needs to block paths from intervened variables to their descendants, and in turn, $z$ would need to be (an invertible transform of) the intervened variables themselves; I cannot think of other scenarios for this to happen.
- **Lemma 1**: The proof is incoherent and circular, which invalidates most of the theoretical analysis thereafter. The "two equivalent models" introduced are identical; Markov equivalence classes (MECs) are irrelevant here. The claim that "they represent the same model" (presumably referring to the posteriors) is tautological, given the models' equality, and is unrelated to Assumption 2. The assertion that "the two resulting distributions over $V$ must be the same" lacks any logical connection to prior statements, and none of Assumptions 1–3 imply equal distributions. The proof resembles "plausible nonsense," *potentially indicative of LLM-generated content*, warranting further review. Some other parts of the paper share a similar feel as can be seen in this review, although it is less apparent.

**Unconvincing Justification for Zero-Shot Transfer to Unknown Causal Settings**

- **Methodological Issues**: How does the approach generalize outside of "a pre-defined class of causal models" $\mathcal{M}$? Defining a proper $\mathcal{M}$ is nontrivial, as identifiability (per Assumption 1) depends not just on graphs but on conditions like positivity (a.k.a overlap) and monotonicity (a kind of functional form). Does $\mathcal{M}$ include only models identifiable from graphs alone? To date, this is limited to unconfounded settings satisfying the backdoor criterion. Even within $\mathcal{M}$, different functional forms would prevent generalization.
- **Experimental Issues**: Results rely on a synthetic linear additive model and a semi-synthetic dataset, which are too narrow to demonstrate robustness. This fails to convincingly show that the method "successfully recovers causal information at inference time even on novel instances" or enables "zero-shot transfer without knowing the causal relations."
- Relatedly, I am not sure how the “model can make inferences on datasets coming from the class of predefined causal models, even if the specific dataset was not in the training set.” How about different models generate highly non-overlapping data? This is not just a theoretical problem, because overlap conditions are inherently required for causal inference [3].

The emphasis on amortized inference as a key feature is underdeveloped. Phrases like "amortization across problem instances" and "amortize over datasets from different causal models" seem to merely describe variational inference with a mixture prior, without clarifying unique benefits in this causal context.

**Confusing Notation and Symbolism**
Notation hampers readability and precision. For example:
- The definition of $D^{\mathrm{tr}}_j$ is unclear:
   - since a single intervention is fixed across models and data, $\mathrm{do}(V)$ should not appear in $D^{\mathrm{tr}}_j$.
   - The symbol $v^{M_{\mathrm{do}(V)}}$ is ambiguous—why a single $n$? It likely intends a dataset, so define $D^{M_{\mathrm{do}(V)}} := \\{v_k^{M_{\mathrm{do}(V)}}\\}_{k=1}^K$.
   - Since $j$ indexes models, use $M_j$ inside $D^{\mathrm{tr}}_j$ consistently, not generic $M$.
- Line 215: Symbol $O$ is used without definition, which ties into the unsubstantiated zero-shot claims.

**Other Issues**
- Assumption 4: The posterior should be absolutely continuous with respect to the prior (reverse the direction). Regardless, this is trivial and redundant: the posterior is a Dirac delta, and the prior is a mixture of Diracs. A real issue is why "absolute continuity implies prior knowledge on which latents do not fall into the observational equivalence class of the input dataset."
- Line 66: Claiming one's own work offers "rare fundamental insights" is oddly self-congratulatory.

**Missing (Proper Discussion of) Related Work**
The paper overlooks key connections to identifiable VAEs and causal applications. Assumption 3's *injectivity* aligns with identifiable VAEs [1], extended to causal effects [2], which also addresses *prior/posterior degeneration to Diracs*.  And I cannot see how (Louizos et al., 2017; Vowels et al., 2021; Wu & Fukumizu, 2023 [2]) are "without modeling the whole distribution".

---

### References
- [1] Khemakhem, Ilyes, et al. "Variational autoencoders and nonlinear ICA: A unifying framework." *International Conference on Artificial Intelligence and Statistics*. PMLR, 2020.
- [2] Wu, Pengzhou Abel, and Kenji Fukumizu. "$\beta$-Intact-VAE: Identifying and Estimating Causal Effects under Limited Overlap." *International Conference on Learning Representations* (2022).
- [3] Rosenbaum, Paul R. "Modern algorithms for matching in observational studies." *Annual Review of Statistics and Its Application* 7.1 (2020): 143-176.

**Questions:**

Please refer to the points in Weaknesses.

---

> ### Author Response · Authors · 2025-11-21
>
> We thank the reviewer for their feedback and the time dedicated to an in-depth review of our work.
>
> - Regarding Assumption 1, we appreciate the reviewer’s suggestions on possible relaxations. For our current setting, the stated form is appropriate, but we agree that weaker identifiability conditions could broaden the theoretical scope. Importantly, even if Assumption 1 is stricter than necessary, this does not undermine the soundness of our approach; if future work confirms that the assumption can be relaxed, our results would only strengthen. For Assumption 2, we would appreciate clarification on the “central problem” the reviewer believes we are sidestepping. We do not claim to establish new identifiability results; our goal is to characterize the distributions ACTIVA estimates under stated assumptions. We agree that posterior parameter identifiability remains an active research area, which is precisely why we present Assumption 2 as an assumption rather than a derived result.
> Regarding Equation 4, we believe the concern stems from a misunderstanding of our setup. The statement that $z$ contains all necessary information follows directly from Assumption 2 and Lemma 1: if $q$ identifies the causal model and is a Dirac distribution, then $z$ encodes the full causal model, which is sufficient to reconstruct the data. Furthermore, the reviewer correctly notes that variables are dependent under interventions. This is also true in our framework; the key modeling choice is that they become conditionally independent *given* $z$. Thus, our conditional independence assumption does not contradict the intervention-induced dependencies. Lastly, we agree that the correct interpretation of z would be where it is as least as informative as the intervened variables, pending in-depth analysis.
>
> - Regarding the critique of Lemma 1: we believe part of the concern arises from a misreading of “two equivalent models.” As defined in the background section, two models are equivalent when they share the same parameters; they are therefore identical. Markov equivalence plays no role in our argument, and we do not suggest otherwise.  Furthermore, we argue that the two samples from the posterior “representing the same model” is not tautological or unrelated to Assumption 2. Rather, it restates the assumption explicitly to make the proof steps transparent. For completeness, we can expand the explanation of why equivalent models induce identical observational and interventional distributions, which we considered self-evident. We address the remark on “plausible nonsense” separately at the end of this rebuttal.
>
> - Addressing the methodological concerns: we agree that specifying an appropriate class of models for training is non-trivial and may limit applicability in some settings. However, two points mitigate this issue.  1) In our experiments, we observe that even when the training class is not identifiable (e.g., the Gaussian case), the method still produces interventional estimates of sufficient accuracy.  2) A key advantage of our approach is that it does not require an explicit parametric specification of the causal models generating the training data; simulators can be used directly. This substantially broadens the range of problems for which the method can be applied. Overall, we argue that full identifiability of the training model class is not strictly necessary for practical success.
>
> - Regarding your experimental issues: for the datasets we evaluated, the results support our claims, particularly with respect to average performance measures. We do acknowledge though, that our experiments are rather minimal examples that aim at validating our framework. Trivially, more experiments on different kinds of data would provide stronger evidence.
> Thank you for the suggestions regarding phrasing and presentation. We agree that the motivation for amortized inference in causal settings can be emphasized more clearly, and we will revise the text accordingly. We also appreciate the notation-related feedback; managing notation across multiple abstraction layers and dimensions is indeed challenging. For completeness, $O$ denotes the Markov equivalence classes (see Appendix A.1).
>
> - Regarding the related works, we appreciate the pointers and will try to incorporate them in appropriate places and add nuance to the other related work references.
>
> We sincerely hope that these detailed clarifications successfully address your concerns and could convince you of the merits of our work.

---

> > ### Author Response · Authors · 2025-11-21
> >
> > Regarding the “plausible nonsense” remark: we found this comment and the projection discouraging and not conducive to a constructive scientific exchange. This is particularly unfortunate given that the comment appears to stem from a misreading of our work (while even if it wasn’t, still unjustified). Substantial effort and care went into this manuscript, and we hope this is acknowledged in the review process. We were also greatly concerned by the suggestion that parts of the submission may have been generated by an LLM. Current ICLR guidelines explicitly allow the use of LLMs with proper disclosure, which we provided, and we view such tools as increasingly important and enabling in ML research. Moreover, the proof in question was developed through human thought process and effort , and not generated by any LLM. In this context, the remark has a similar tone to the “plausible nonsense” comment. We respectfully ask that future discussion remain focused on the scientific content, and we welcome any concrete, substantive feedback that can help improve our work.

---

### Official Review · Reviewer_U8Ab · 2025-11-10

**Soundness:** 1
**Presentation:** 3
**Contribution:** 3
**Rating:** 2
**Confidence:** 3

**Summary:**

This paper proposes an algorithm for zero-shot prediction of interventional distributions given observational data as input. For this purpose, they utilize a transformer-based conditional VAE and use both observational and interventional data along with the corresponding causal query as input during training. They provide the theoretical insight behind their capability of amortized causal inference. Finally, the authors provide experimental results on synthetic and semi-synthetic datasets.

**Strengths:**

The paper proposes a solution to a very interesting and important problem. It is also written in a very intuitive way. I appreciate how they precisely mentioned the assumptions and wrote the proofs for their theoretical statements.

**Weaknesses:**

Below I share my comments:

## **Major:**

1. It is unclear what the proposed method is generalizing over. Can it generalize to
   i) arbitrary observational distributions,  ii) arbitrary causal graphs,     iii) arbitrary interventions?

2. **(i) Arbitrary observational distributions:**  What if the model was trained on observational datasets $D^M$ with noises being only Gaussian, however, during inference, the observational datasets $D^M$ have different noise distributions?

3. **(ii) Arbitrary causal graphs:**  Does the method not need any assumption on the graph $G$? In line 461, the authors mentioned that the need for graphical assumptions is removed.
   We know that based on the causal graph, the causal effect might be different for the same query. How does the model deal with such a scenario?

   Suppose, for a 3-node graph of $X, Z, Y$, the model was not trained on any dataset which represented the $X \rightarrow Z$ edge, but during inference time we give a dataset as input which has $X \rightarrow Z$ dependency. The model should not be able to generalize in such a scenario unless the model was trained on datasets generated from all possible causal graphs, which would be exponentially costly.

4. **(iii) Arbitrary interventions:**  In a causal model with $d$ variables, $d$ single interventions are possible. Do they consider $d$ interventional datasets?  Do the authors not need any assumption on how many interventions they are considering?  How does the model generalize to higher support size?

   More precisely, suppose during training the model was given interventional datasets with $\text{do}(X=0), \ldots, \text{do}(X=8)$, but during inference, we give a query $\text{do}(X=9)$. Will the model generalize in such a scenario?

   Also, suppose during training, we gave it $\text{do}(V_1), \text{do}(V_2), \ldots, \text{do}(V_{n-1})$, i.e., $n-1$ different interventional datasets. During inference, we query $\text{do}(V_n)$ or $\text{do}(V_2, V_3, V_4, V_5)$, i.e., interventions on a new set of variables unseen in the training data. It should not generalize, as interventions/treatments on multiple variables might interact with each other and give a different output, unless the model was trained on all possible interventional datasets.

5. According to lines 261 and 272, for just one combination of intervention (e.g., $\text{do}(X=0)$), the authors need a dataset of size $N \times d \times |I| + 1$.   What interventional datasets does the algorithm need as input for each of the above cases?  I understand we need at least the observational data $D$ sampled from $P(\mathbf{V})$.

6. More baselines are needed. The authors compared with the conditional model baseline. We understand that the method performs better than the baseline, but are the errors obtained by the algorithm low enough to be considered acceptable? This is not clear.
   They should compare their performance with existing algorithms for causal effect estimators that take the observational data, the causal graph, the query and gives a causal effect prediction. For example: Shpitser, Ilya, and Judea Pearl. "Complete identification methods for the causal hierarchy." (2008).

---

## **Minor:**

1. This paper considers no unobserved confounder, i.e., all exogenous noises are independent. This needs to be stated more precisely.

2. Some end-to-end diagram of the algorithm would be helpful for readers to connect the whole algorithm. For example, how are the $\lambda$, $\phi$, and $\eta$ parameterized models arranged?

**Questions:**

Below I share my questions:

1. How does the proposed approach work for multiple interventions during inference?
2. For two variables case, conditioning and intervening should be the same. In Table 1, why is the error so high for the conditional baseline in the two-variable case?
3. How do the 2-variable ($X \rightarrow Y$ or $Y \rightarrow X$) and 8-variable graphs look like?
4. Is there any possibility that the model $q_{\phi}$ just ignores $\mathbf{V}$? When might that happen?

---

> ### Author Response · Authors · 2025-11-21
>
> We thank the reviewer for their time and in-depth treatment of our work. Your review sparked interesting insights into our work and will help improve future work further. In the following we address your concerns and questions:
>
> - (i) Arbitrary Observational Distributions: (i) Our amortized inference approach relies on the synthetic training data being drawn from the same distribution of SCMs as the data observed at inference time (lines 159–161). When the inference-time SCM is out of distribution, degraded performance is expected. We explicitly evaluate this behavior in the semi-synthetic experiments, where the test data are OOD.
>
> - (ii) The causal graph is not required during inference. This is a central contribution of the paper. The reviewer is correct that inference may fail when the inference-time SCM is not represented in the training distribution, but:
> 1. As long as the full SCMs at inference are sampled from the same distribution as those used for training—and assuming sufficient training coverage—our method is expected to generalize (lines 153–161). This applies to the entire SCM, not only the graph. Specifically, we expect our method to estimate a similar mixture of distributions as for the observationally equivalent models in the training set.
> 2. The concern about unseen graph structures is mitigated by the permutation-equivariant architecture. If the model has learned to infer the mixture of interventional distributions for a structure such as $X \rightarrow Y \rightarrow Z$, it can infer the corresponding distribution for any relabeling of this graph (e.g., $Y \rightarrow X \rightarrow Z$), provided the underlying functional parameters match. Thus, even if a specific edge orientation such as $X \rightarrow Z$ did not appear during training, the model can still perform inference when an isomorphic graph—under a node permutation—was observed. For DAGs with $n$ nodes, permutation equivariance reduces the number of distinct graphs the model must encounter by roughly a factor of $n!$, substantially improving coverage of the space of possible structures.
>
> - (iii) We distinguish two types of intervention generalization:  1) New intervention values on previously intervened variables, and 2) Interventions on variables (or variable sets) not intervened on during training.
> 1) For unseen intervention values on variables that were intervened upon during training, our setting is analogous to standard ML generalization. If a model has learned the effect of an intervention at certain values, its ability to generalize to new values depends on whether the mapping varies smoothly with respect to those values. Although we did not evaluate this explicitly, we expect a certain degree of  generalization.
> 2) For interventions on entirely new targets, we agree that limitations arise: the model has not learned the mapping between those targets and the corresponding equivalence class of interventional distributions. In practice, however, the set of likely intervention targets is often known in advance. In the semi-synthetic experiment, for example, one might anticipate gene  knockouts for at most $n$ variables. Under permutation equivariance, observing all combinations of these $n$ targets is unnecessary, because structurally identical intervention scenarios under node relabeling are already captured. Again, this reduces the required training coverage, substantially.
>
> - 5. The size of the training input does not scale with the number of possible intervention targets, but only with the number of possible intervention *values*. The additional tensor dimension $(|I| + 1)$ corresponds to a one-hot encoding that indicates which intervention value (if any) was applied to each variable in each sample. For example, if a sample has observed value $v_{1,4,0}$, then setting $v_{1,4,3} = 1$ specifies that variable 4 in sample 1 received the third intervention value from the predefined set $|I|$. Thus, adding new intervention *values* increases the size of this encoding dimension by one, whereas adding new potential intervention *targets* does not affect input dimensionality.
>
> - 6. Additional baselines would indeed be valuable, and we have explored methods we could compare to.. However, we are not aware of existing methods that address the same task under comparable assumptions. Available approaches typically rely on substantially stronger side information (e.g., known graphs or covariates), target only specific causal quantities such as ATEs rather than full interventional distributions, or require non-amortized retraining for each individual instance. As a result, performance comparisons would be difficult to interpret and likely uninformative.
>
> - Thank you also for the suggestions in the minor comments. We aim to address them in a potential camera ready version of this paper.

---

> ### Author Response · Authors · 2025-11-21
>
> Regarding your questions:
> 1. For multi-target interventions, the intervention-value vector is colocated with each intervened variable. See also the construction described in point 5 above.
>
> 2.  The conditional baseline fails because the intervened variable is not necessarily a cause. The baseline fits a Gaussian to the observational joint distribution $p(X, Y)$ and answers every query using the corresponding conditional. When the queried variable is actually an effect, the correct interventional distribution on the cause is its marginal $p(\text{cause})$. The baseline instead uses $p(\text{cause} \mid \text{effect} = v)$, which reflects observational correlation rather than causal structure, leading to incorrect estimates.
>
> 3. In all experiments, graphs are generated by randomly sampling edges. Details are provided in Appendix A.2.1.
>
> 4. The intent of the question is unclear. Clarification would help us provide a precise response. Do you mean that the model ignores (i) the observed samples $V$, or (ii) the intervention specification $\mathrm{do}(V)$?
>
> We hope these clarifications resolve the concerns raised and provide a clearer view of the soundness and contribution of our approach.

---

### Meta-Review · Area_Chair_n6cQ · 2026-01-04

**Summary:**

The reviewers are concerned about novelty, the soundness of theoretical claims, and assumptions about this paper. I agree with most of these concerns. Completely sidestepping causal graphs can only be done under very specialized settings and not in the general way advocated by the authors. Reviewer U8Ab raised a good point about generalizability and the authors' rebuttal emphasizes the issue with the method: They need the SCM to be within the class of trained SCMs. This is typically not possible.

**Reviewer Concerns:**

I do not believe any of these concerns are successfully addressed by the rebuttal.

**Reviewer Scores:**

My guess is that the extremely supportive reviewer who is not confident would have reduced their score after reading the other reviews pointing out to the issues they missed.

---

### Decision · Program_Chairs · 2026-01-26

Reject